# Determinants of institutional delivery service utilization in Nepal

Bipin Thapa[1]*, Anita Karki[2], Suman Sapkota[3], Yifei Hu[3]

1 Department of Research and Development, Dhulikhel Hospital-Kathmandu University Hospital, Kavre, Nepal, 2 Central Department of Public Health, Institute of Medicine, Tribhuvan University, Kathmandu, Nepal, 3 Department of Child and Adolescent Health and Maternal Care, School of Public Health, Capital Medical University, Beijing, China

* bipinthapa2050@gmail.com

## Abstract

### Background

Maternal mortality continues to be a pressing concern in global health, presenting an enduring and unmet challenge for healthcare systems worldwide. Utilization of institutional delivery services has been established as a proven intervention to mitigate life-threatening risks for both mothers and newborns. Exploring the determinants of institutional delivery is crucial to improve and enhance maternal and newborn safety. This study aimed to assess the contextual and individual factors associated with institutional delivery in Nepal.

### Methods

This study utilized that data form Nepal Multiple Indicator Survey 2019, which included a sample of 1,932 women who had given birth within the two years prior to the survey. A multilevel logistic regression analysis was performed to determine the significant external environment, contextual and individual predictors of institutional delivery.

### Results

The women from Madhesh province [Adjusted Odds Ratio (aOR): 0.32, 95% Confidence Interval (CI): 0.17–0.61], as compared to Bagmati province, women from rural areas (aOR: 0.55, 95% CI: 0.39–0.78) as compared to urban areas, and women from a relatively less-advantaged ethnic groups (aOR: 0.52, 95% CI: 0.35–0.76) as compared to the relatively advantaged ethnic groups were less likely to deliver in health institutions. Similarly, women from the poorest (aOR: 0.09, 95% CI: 0.04–0.22) and second wealth groups (aOR: 0.29, 95% CI: 0.13–0.64) were less likely to attend institute for delivery compared to women from the richest household. Women with formal education (aOR: 1.65, 95% CI: 1.16–2.35) were more likely to deliver in an institution over uneducated women. Moreover, the uptake of institutional delivery increased by 59% (aOR: 1.59, 95% CI: 1.43–1.75) for each additional ANC visit.

**Data Availability Statement:** Data are available in a public, open access repository. We used publicly available data and it is accessible from the MICS website (https://mics.unicef.org/surveys) on the request.

**Funding:** The author(s) received no specific funding for this work.

**Competing interests:** The authors have declared that no competing interests exist.

## Conclusion

The findings highlight the importance of stepping up efforts to achieve universal health care from the standpoint of long-term government investment, focusing particularly on illiterate women in rural areas, poorer households, and socially disadvantaged groups. Expanding the benefits of maternal benefit schemes targeting the women from the poorest households in the communities is recommended.

## Background

Maternal mortality remains a global problem. Almost 810 women worldwide die from complications related to pregnancy and childbirth each day [1]. Alarmingly, almost nineteen out of every twenty (94%) maternal deaths occur in low-resource settings while Southern Asia alone accounts for nearly one-fifth of global maternal deaths [1]. Maternal mortality has decreased globally over the past decade, nevertheless, significant inequalities remain to persist both within and between countries [1]. Consequently, addressing maternal mortality remains a critical concern in the global health agenda, posing an ongoing and unmet challenge for healthcare systems worldwide.

Institutional delivery service utilization is one of the proven interventions to lower health risks for both mothers and newborns. Institutional delivery services effectively minimize complications and prevent infections, thereby increasing the survival rate of mothers and newborns. Delivering at healthcare facilities enables women to receive proper medical attention and care during childbirth. This is essentially advocated as the most effective method of avoiding mortality of pregnant women and newborns [2]. Globally, approximately three-fourth (76%) of deliveries were conducted in healthcare facilities in 2015 [3] while global maternal mortality ratio (MMR) was reported to be 216 per 100000 live births (LB) [4]. In Nepal, however, the 60.2% of the deliveries were conducted in health institutions which is lower than the global average. Moreover, Nepal's maternal mortality ratio (MMR) stood at 258 per 100,000 live births (LB), surpassing the global average. In fact, Nepal's MMR was higher than its South Asian neighbors namely India (174), Bhutan (148), Bangladesh (176), Myanmar (178), Pakistan (178), and Sri Lanka (30) [5].

Nepal has made commendable progress in reducing its MMR, decreasing from 850 deaths per 100,000 LB in 1990 to 258 deaths per 100,000 LB in 2015. Despite this improvement, Nepal was only able to partially achieve the Millennium Development Goal (MDG) target of 213 per 100,000 LB [6]. Impact of national programs like Aama Surakshya (Mother's Safety), which promotes four antenatal visits and institutional delivery, and the Birth Preparedness Package (BPP) in promoting institutional delivery has been momentous [7]; the proportion of women who delivered in health facilities increased from 9% in 2001 to 77.5% in 2019 [8, 9]. Currently, the government of Nepal has further committed to reduce MMR at 70 per 100,000 LB by 2030 to achieve sustainable development goals (SDG) [10]. The latest estimates of MMR, according to the report on maternal mortality by national population and housing census 2021 is 151 per 100,000 livebirths which lags behind the SDG target of 116 per 100,000 LB by 2022 [10]. In this pace, achieving SDG is a challenging task with a reduction of 73% in MMR, which is considerably higher than the preceding period's reduction of 53% (2000–2015) [4]. This highlights the need for Nepal to accelerate the progress in the uptake of maternal health services, along with institutional delivery to meet the MMR targets of SDG. Moreover, the national averages mask large disparities among women from various geographic and socio-economic origins utilizing institutional delivery services [9, 11]. Thus, it is crucial to identify and address the disparities

in the utilization of maternal health services, including institutional delivery services, across various groups.

A limited number of studies have used national survey data to determine factors influencing the utilization of institutional delivery services. Notably, only one study conducted by Neupane et al. used the data from Nepal demographic and health survey (NDHS) 2016 to determine the external environmental, predisposing, and enabling factors associated with the use of institutional delivery care in Nepal using multilevel regression analysis [12]. Therefore, this study aims to fill the research gap by assessing the contextual and individual factors associated with institutional delivery in Nepal using the recent data from Nepal Multiple Indicator Cluster Survey (MICS) 2019. This study employs multilevel binary logistic regression analysis to determine the difference in uptake of institutional delivery at regional, household, and individual level and provides a comprehensive understanding of the factors influencing institutional delivery uptake in Nepal.

## Material and methods

### Data sources

The present study was based on data from the MICS 2019 conducted by the Central Bureau of Statistics (CBS) from May to November 2019 in collaboration with the United Nations Children's Fund (UNICEF). The survey applied a multistage (2-stage) sampling technique. The urban and rural areas within each province were made the main sampling strata. Kathmandu valley urban was included as a separate stratum. Within each stratum, primary sampling units (PSUs) were selected systematically with probability proportional to size. Furthermore, a household listing was carried out within the selected PSUs, identifying the households with and without children under five years. In total, 25 households with and without children under five were selected in each PSU through a systematic random sampling method. The households with children under five were oversampled for the assurance of strong representativeness of children under five in the sample, where 13 households with children under five and 12 households without were selected from the listing in each Enumeration Areas (EAs). The detailed methodology of the survey can be accessed from the Nepal MICS 2019 report [9]. The website of the MICS program makes the anonymized survey dataset freely available under request [13].

### Participants and sample size

A total sample of 512 PSUs and 12,800 households was selected for the survey where 12,655 were interviewed. A total of 14,805 women aged 15–49 years were interviewed which included 6,658 mothers/caretakers of children under five years. After merging two different data sets i.e., mother and household, a total of 14,805 observations were found. 1,950 observations of married women aged 15–49 years with a live birth in the last two years by place of delivery of the most recent live birth were selected and retrieved. After removing five observations with missing values: the place of delivery (n = 1), health insurance (n = 3) and household head education (n = 1), a total of 1,945 observations remained. Finally, 13 observations which did not fit the criteria for the place of delivery were removed and 1,932 observations were considered in the final analysis.

### Conceptual framework

We categorized the determinants of institutional delivery into external environment, household contextual and household individual groups based on extensive literature review and expert opinion. The conceptual framework for the study is shown in Fig 1.

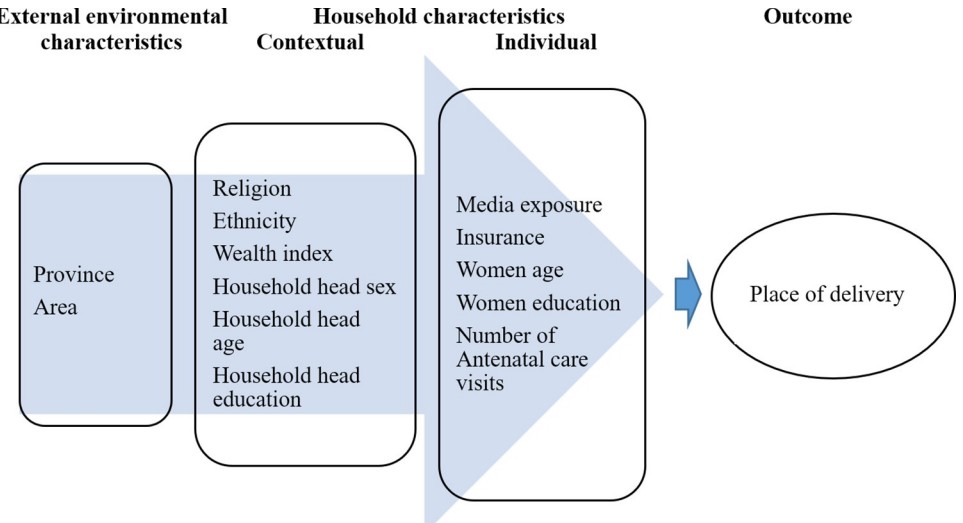

**Fig 1. Conceptual framework for institutional delivery service utilization in Nepal.**

## Outcome variable

Our outcome variable was the place of delivery, that is, institutional delivery or home delivery. Delivery in the government hospital, government clinic/health centre, health post, private hospital, private clinic and maternity home or private medical institute was considered institutional delivery, whereas delivery in the respondent's home or other's home was considered home delivery.

## Covariates

Altogether 13 characteristics relevant to the study were selected and divided into three different categories. External environment variables included the province of residence and area of residence. Household contextual variables included religion, ethnicity, household wealth index quintile, household head's age, household head's sex and household head's educational status. Likewise, household individual variables included women's media exposure, enrollment into health insurance, women's age, women's educational status and antenatal care (ANC) visits. The description and measurements of the explanatory variables are presented in Table 1.

## Statistical methods

### Data management and analysis

After approval of the request by UNICEF MICS team, data was downloaded from MICS website survey page and received in Statistical Package for Social Science version 23 format. Two datasets namely the household dataset and the women dataset were used for this study. Required variables from household were merged into the women's dataset. The variables were then created and recoded according to the objectives of the study using Statistical Analysis System OnDemand for Academics and syntax and output files were well documented.

Descriptive statistics were reported for all the variables. Women weight, cluster and strata were adjusted to calculate the descriptive statistics. At first, bivariate analyses were conducted where an unadjusted odds ratio with a 95% confidence interval was calculated, and variables significant at P<0.05 in bivariate analysis were considered as candidate variables for the multilevel analysis. Furthermore, the variables were categorized as external environment factors,

**Table 1. Description and measurements of study variables.**

| Variables | Definition of variables | Measurements |
|---|---|---|
| **External environment** | | |
| **Province** | Province where the household is located | Province 1, Madhesh province, Bagmati province, Gandaki province, Lumbini province, Karnali province, Sudoorpaschim province |
| **Area** | Level of urbanization | Urban (metropolitan city, sub-metropolitan city, municipality), Rural: Rural municipality |
| **Household contextual factors** | | |
| **Religion** | Religion of the household head | Hindu, Non-Hindu (Buddhist, Islam, Kirat, Christian, Prakriti) |
| **Ethnicity** | Ethnicity of household head | Relatively advantaged (Brahmin, Chhetri, Gurung, Magar, Newar, Thakuri), Relatively disadvantaged: Others |
| **Wealth index quintile** | A composite score was generated by principal component analysis using information on consumer goods ownership, dwelling characteristics, water and sanitation, and other assets and durables related to household wealth. Survey households were divided into equal five parts according to their wealth score | Poorest, Second, Middle, Fourth, Richest |
| **Household head's age** | Age of household head in completed years | 17 to 95 years (continuous form) |
| **Household head's sex** | Sex of household head | Male, female |
| **Household head's education** | Educational status of household head | No education, Formal education |
| **Household individual factors** | | |
| **Media exposure** | Women's exposure to television, radio and newspaper | Adequate exposure (exposure to television or radio or newspaper at least once a week), Inadequate exposure (exposure to television or radio or newspaper less than once a week or not exposed at all) |
| **Health insurance** | Women covered with health insurance | With insurance, Without insurance |
| **Women's age** | Age of women in completed years | 15 to 49 years (continuous form) |
| **Women's education** | Educational status of women | No education, Formal education |
| **Number of ANC visits** | Total number of times women received prenatal care prior to child delivery | Number of ANC visits (continuous form) |

contextual factors and individual factors for further analysis in multilevel regression. The multilevel nested structure of the analysis comprised 1,932 individuals grouped (level 1) nested into 504 PSU (level 2). The estimation of generalized linear mixed models, including random effects and model fitness, was carried out using the GLIMMIX procedure. Weight was adjusted and the cluster was placed at the subject statement. A parsimonious final model for institutional delivery was obtained where the significant variables in the previous models were carried over to the subsequent models as per the conceptual framework (Fig 1); predictors that remained statistically significant at 5% (P ⬚ .05) were retained in the analysis for adjustment in the next model. The first model consisted of external environment factors. Furthermore, household contextual factors were added in the second model, while individual factors were added in the third model. The covariance parameters were significant for all the models. The multicollinearity of the variables was tested before entering them into the models. No multicollinearity was observed among the independent variables. Adjusted odds ratios (aOR) with 95% confidence intervals (CI) were calculated to identify the variables with a higher likelihood of institutional delivery.

## Ethical considerations

The survey protocol for MICS was approved by the Central Bureau of Statistics (CBS) as per the Statistical Act (1958) in September 2018. The Statistical Act enables CBS to carry out

surveys according to the government's ethics protocol without involving an institutional review board. Questionnaires were administered with comprehensive introductions and obtained verbal consent from respondents. Participants were explicitly informed of the survey's voluntary nature, their right to decline answering specific questions or discontinuing the interview at any point, and the assurance of confidentiality and anonymity for their information.

# Results

## Socio-economic and demographic characteristics of respondents

The socio-economic and demographic characteristics of respondents are presented in Table 2. The percentage of women attending institutional delivery was lower in Karnali and Madhesh

**Table 2. Socio-economic and demographic characteristics of respondents.**

| Characteristics | | Institutional delivery n(%) | Odds ratio (95% CI) |
|---|---|---|---|
| Province | Province 1 | 239(79.1) | 0.39(0.20–0.76)[b] |
| | Madhesh | 267(64.6) | 0.14(0.07–0.26)[a] |
| | Gandaki | 137(91.4) | 1.28(0.53–3.11) |
| | Lumbini | 289(78.6) | 0.40(0.21–0.77)[b] |
| | Karnali | 82(62.9) | 0.15(0.07–0.31)[a] |
| | Sudoorpaschim | 157(84.5) | 0.62(0.29–1.33) |
| | Bagmati | 340(89.2) | Ref |
| Area | Rural | 446(66.9) | 0.27(0.18–0.40)[a] |
| | Urban | 1067(84.2) | Ref |
| Religion | Non-Hindu | 220(71.2) | 0.63(0.42–0.95)[c] |
| | Hindu | 1293(79.6) | Ref |
| Ethnicity | Relatively less advantaged group | 862(72.5) | 0.32(0.22–0.45)[a] |
| | Relatively advantaged group | 650(87.5) | Ref |
| Wealth index quintile | Poorest | 252(57.7) | 0.03(0.01–0.07)[a] |
| | Second | 301(74) | 0.08(0.03–0.17)[a] |
| | Middle | 310(80.9) | 0.12(0.05–0.28)[a] |
| | Fourth | 336(87.9) | 0.24(0.11–0.54)[a] |
| | Richest | 313(96.9) | Ref |
| Household head's age | Median (P25, P75) | 47(31,58) | 1.01(0.99–1.01) |
| Household head's sex | Male | 1165(76.9) | 0.73(0.52–1.04) |
| | Female | 348(83.3) | Ref |
| Household head's education | Formal education | 961(82.5) | 1.59(1.20–2.10)[b] |
| | No education | 551(71.8) | Ref |
| Media exposure | Adequate | 1026(85.7) | 2.76(2.07–3.67)[a] |
| | Inadequate | 486(66.2) | Ref |
| Health insurance | With insurance | 88(94.7) | 4.10(1.50–11.19)[b] |
| | Without insurance | 1424(77.4) | Ref |
| Women's age | Median (P25, P75) | 25(21, 29) | 0.96(0.94–0.98)[b] |
| Women's education | Formal education | 1294(84.6) | 4.56(3.34–6.22)[a] |
| | No education | 219(54.4) | Ref |
| Number of ANC visits | Median (P25, P75) | 4(4, 5) | 1.89(1.71–2.09)[a] |

Note: [a]p<0.001

[b]P<0.01

[c]P<0.05, CI: Confidence Interval, P25: 25th percentile, P75: 75th Percentile

provinces, namely 62.9% and 64.6%, respectively, while it was highest in Bagmati province (89.2%). Slightly over 84% of women had institutional delivery in urban areas, compared to 66.9% of women in rural areas. Both external environmental factors, such as province and area of residence, were significantly associated with the place of delivery.

For household contextual factors, approximately 80% of Hindus and 70% of non-Hindus delivered in institutions. Similarly, the percentage of institutional delivery was higher among women from relatively advantaged ethnic groups, at 87.5%, while only 72.5% of women from relatively less advantaged ethnic groups delivered in health facilities.

About 58% of women from the poorest wealth quintile delivered in institutions, while the percentage gradually increased with each wealth quintile. Almost 97% of women from the richest wealth quintile delivered in health facilities. The median age of the household head was 47 years for women who had health institution-based delivery. About 83% of women whose household head's sex was female delivered in health facilities, whereas the percentage was lower among women whose household head's sex was male (76.9%). The percentage of women attending institutional delivery was approximately 83% when the household head had received formal education, but it was 71.8% among women whose household head was uneducated. Religion, ethnicity, wealth quintile, and household head's education showed a significant association with the place of delivery.

Likewise, for individual factors, almost 86% of the women who had adequate media exposure delivered their babies in health facilities, whereas about nearly 66% of the women who didn't have adequate media exposure attended health institution-based delivery. Nearly 95% of the women who were covered with insurance had institutional delivery, however, the percentage was lower for the women who were not covered by health insurance (77.4%). Both media exposure and insurance were significantly associated with the place of delivery. The median age of women attending institutional delivery and home delivery was 25 years each. Regarding education, 84.6% of women who had attended formal education had institutional delivery, whereas only more than half of the women who were uneducated delivered in health facilities. The median number of ANC visits for women delivering in health facilities was four, whereas it was three for women delivering at home. Women's age, women's education and the number of ANC visits were significantly associated with the place of delivery. The percentage of women delivering in health institutions was 78.3% (95% CI: 75.3–81.1) while it was 21.7% (95% CI: 18.8–24.6) for home delivery.

## Multilevel logistic regression between the place of delivery and contextual and individual factors

The result of the multilevel logistic regression model for the place of delivery is presented in Table 3. A parsimonious final model for institutional delivery was obtained by carrying the significant variables in the previous model to adjust in the subsequent model. External environment factors were fitted in the first model, while household contextual factors and individual factors were added in the second and the third models respectively.

In the first model, Province 1, Madhesh Province, Lumbini Province, Karnali Province and rural area were negatively associated with institutional delivery. Likewise, in the second model, Madhesh Province, rural area, non-Hindu religion, relatively less-advantaged ethnicity, and poor wealth groups were negatively associated with institutional delivery. In the final model (third), Madhesh Province, rural area, relatively less-advantaged ethnicity, and poor wealth groups were negatively associated with institutional delivery, while women education, and the number of ANC visits were positively associated with institutional delivery.

**Table 3. Multilevel logistic regression between the place of delivery with contextual and individual factors.**

| Factors | Model 1 aOR(95% CI) | Model 2 aOR(95% CI) | Model 3[d] aOR(95% CI) |
|---|---|---|---|
| **Province** | | | |
| Province 1 | 0.47(0.25–0.91)[c] | 0.91(0.48–1.70) | 0.91(0.48–1.71) |
| Madhesh | 0.15(0.08–0.28)[a] | 0.18(0.09–0.33)[a] | 0.32(0.17–0.61)[a] |
| Gandaki | 1.46(0.61–3.46) | 1.38(0.58–3.26) | 1.24(0.52–2.93) |
| Lumbini | 0.51(0.27–0.96)[c] | 0.62(0.33–1.14) | 0.74(0.40–1.37) |
| Karnali | 0.2(0.09–0.41)[a] | 0.68(0.32–1.41) | 0.82(0.39–1.72) |
| Sudoorpaschim | 0.76(0.36–1.57) | 1.47(0.70–3.08) | 1.71(0.81–3.60) |
| Bagmati | Ref | Ref | Ref |
| **Area** | | | |
| Rural | 0.29(0.20–0.42)[a] | 0.54(0.38–0.77)[a] | 0.55(0.39–0.78)[a] |
| Urban | Ref | Ref | Ref |
| **Religion** | | | |
| Non-Hindu | | 0.63(0.42–0.95)[c] | 0.71(0.47–1.08) |
| Hindu | | Ref | Ref |
| **Ethnicity** | | | |
| Relatively less advantaged | | 0.46(0.31–0.67)[a] | 0.52(0.35–0.76)[a] |
| Relatively advantaged | | Ref | Ref |
| **Wealth quintile** | | | |
| Poorest | | 0.04(0.01–0.09)[a] | 0.09(0.04–0.22)[a] |
| Second | | 0.15(0.06–0.33)[a] | 0.29(0.13–0.64)[b] |
| Middle | | 0.27(0.12–0.61)[b] | 0.48(0.21–1.08) |
| Fourth | | 0.44(0.19–0.97)[c] | 0.65(0.28–1.46) |
| Richest | | Ref | Ref |
| **Household head's education** | | | |
| Formal education | | 0.99(0.74–1.31) | - |
| No education | | Ref | |
| **Media exposure** | | | |
| Adequate | | | 1.01(0.74–1.40) |
| Inadequate | | | Ref |
| **Insurance** | | | |
| With insurance | | | 2.39(0.84–6.76) |
| Without insurance | | | Ref |
| **Women's age (continuous)** | | | 0.98(0.95–1.01) |
| **Women's education** | | | |
| Formal education | | | 1.65(1.16–2.35)[b] |
| No education | | | Ref |
| **ANC visits(continuous)** | | | 1.59(1.43–1.75)[a] |
| **Covariance parameter estimate (Intercept)** | 1.32(0.25)[a] | 0.70(0.18)[a] | 0.51(0.17)[b] |
| -2 log-likelihood | 1734.79 | 1587.75 | 1444.94 |
| AIC | 1752.79 | 1619.75 | 1484.94 |

Note: [a]p<0.001

[b]P<0.01

[c]P<0.05

[d]Best fitting model. aOR: Adjusted Odds Ratio, CI: Confidence Interval, ANC: Antenatal Care, AIC: Akaike Information Criterion.

Values based on SAS PROC GLIMMIX, Estimation method = quad

Women from Madhesh province compared to Bagmati province were less likely to deliver in health institution [aOR: 0.32, 95% CI: 0.17–0.61]. Similarly, women residing in rural areas were less likely to attend institutional delivery compared to women form urban areas (aOR: 0.55, 95% CI: 0.39–0.78). Likewise, women from relatively less-advantaged ethnic groups had 48% (aOR: 0.52, 95% CI: 0.35–0.76) lower chances of giving birth in health institution than women from relatively advantaged ethnic group. Moreover, the results showed that, compared to women from the richest household group, women from the poorest (aOR: 0.09, 95% CI: 0.04–0.22) and second wealth group (aOR: 0.29, 95% CI: 0.13–0.64) had a lower likelihood of giving birth in health institution. The odds of institutional delivery increased with the educational level where women with formal education were nearly twice as likely (aOR: 1.65, 95% CI: 1.16–2.35) to deliver in an institution over uneducated women. Moreover, the odds of having an institutional delivery increased by 59% for each additional ANC visit (aOR: 1.59, 95% CI: 1.43–1.75).

## Discussion

This study assessed the contextual and individual factors associated with institutional delivery in Nepal with the latest Nepal Multiple Indicator Cluster Survey 2019 data employing a multi-level modelling approach. Province, area, ethnicity, wealth index, women's education, and the number of ANC visits were significantly associated with institutional delivery.

### Influence of external environment factors on utilization of institutional delivery

Our study found a provincial disparity in the uptake of institutional delivery which corroborates with previous studies from Nepal, which reported uneven progress in closing the equity gap in the use of maternal health services in seven provinces of Nepal [12, 14, 15]. In support, disparities in the utilization of health institution-based delivery according to geographical regions have been reported from various countries as well [16–18]. The provincial differences in the utilization of delivery services in Nepal may be explained by geographical barriers, unavailability of and inaccessibility to health facilities, socio-economic status, and prevailing traditional beliefs about childbirth. In our study, women from Madhesh province were less likely to deliver in a health institution than the women from Bagmati province. Even though Madhesh province is largely plain in terms of topography and therefore has greater physical access to health services than other provinces, the lower likelihood of health institution delivery can be explained by poor socio-economic indicators. In addition, the underlying sociocultural practices and traditional beliefs could also be a major barrier that hinders women from accessing delivery services. The finding from the 2016 Nepal Demographic and Health Survey (NDHS) showed that nearly 80% of mothers in Madhesh province felt unnecessary to deliver in a health institution [19]. Many women consider the birthing process as a part of natural life events and only consider seeking institutional delivery care services only if complications arise [20].

In our study, women from rural areas were significantly less likely to deliver in a health institution. This finding is consistent with previous studies [21, 22]. The rural areas in Nepal are characterized by harsh geographical conditions, lack of transportation facilities, and unavailability of health services and health workers. A qualitative study by Shah et al. cited traditional socio-cultural norms and values disfavoring institutional deliveries, lack of access to birthing facilities, poor perception regarding quality of health services, insufficient financial incentives, and poor infrastructures and equipment at birthing centres as barriers among women from rural areas which deter them from delivering at health facilities [23]. A study

conducted in rural areas of Nepal revealed that the majority of the birthing centres did not meet the required guidelines set by National Safe Motherhood Program (NSMP) and many facilities lacked the basic commodities, medicines and supplies to provide essential ANC, delivery and newborn services [24]. The low levels of availability, readiness, and coverage of obstetric services in rural areas of Nepal could be major reasons for urban-rural differentials in the uptake of institutional delivery services.

## Influence of household contextual factors on utilization of institutional delivery

Several studies have identified caste/ethnicity as one of the important predictors of institutional delivery in Nepal [12, 25–27]. A similar finding has been reported in our study where women from relatively disadvantaged castes were less likely to deliver in a health institution. This could be explained by a social discrimination against women of relatively disadvantaged castes by service providers or other members of the community. A study from Nepal revealed that women's experience of disrespect and abuse during labour and childbirth, was significantly associated with relatively disadvantaged castes [28]. These encounters of mistreatment in the health facilities at any stage in the continuum of care may deter women from seeking maternity care from health facilities and consequently, they might resort to home deliveries. Moreover, lower-caste women are found to be usually deprived of many socioeconomic and educational opportunities as well [29]. The fact that ethnicity has consistently been a factor that influences the uptake of institutional delivery among women suggests that maternal health programs of GoN do not adequately address the preexisting social structures and deeply ingrained cultural issues which are checking women from disadvantaged castes from institutional delivery.

The current study showed that women from poorer household are less likely to deliver in a health institution. Furthermore, the women from poorest households were 91% less likely to deliver in a health institution compared to the women from the richest households which indicates a huge gap in institutional delivery service utilization between the poorest and the richest wealth groups. Differentials in the utilization of delivery services by the household wealth status of women have been evident from previous studies conducted in some South Asian countries as well as Sub–Saharan Africa [30–32]. A recent study by Bhusal also concluded that utilization of institutional delivery was disproportionately higher among women from wealthy groups [33].

GoN introduced the maternity incentive scheme (MIS) in 2005, the Safe Delivery Incentive Scheme (SDIP) in 2006 and the Aama Program in 2009 to reduce the financial barriers to women seeking institutional delivery. Aama Program included free institutional delivery services as well as cash incentives upon completion of four ANC visits [34]. A study reported that institutional delivery increased by five folds after the implementation of the Aama program compared to no free delivery care (FDC) policy [35]. However, a study by Ensor et al. suggested that the beneficial impact of maternal financing policies in Nepal is skewed towards areas and households that are accessible and wealthy [36]. Therefore, the national program which aims to reduce financial barriers may have disproportionately benefited the non-poor household whilst only partly mitigating the financial burden faced by the women from the poorest households. Institutional delivery care costs more than the incentive offered and may not be enough to compensate full range of expenses incurred as a result of delivery which includes transportation costs, opportunity costs, and food and lodging for women from poor households [23, 37]. This existing financial barrier might still be prohibiting the women from the poorest households to seek delivery care services. Moreover, poorer women are more likely

to experience distance-related barriers which further exacerbate their situation and deter them from seeking institutional delivery care services [38].

## Influence of individual modifiable factors on utilization of institutional delivery

This study established women's education as a significant predictor of institutional delivery. This finding aligns with the previous studies [12, 14, 15, 39]. A study by Mehata et al. highlighted that out of all socio-demographic variables, maternal education was the most powerful predictor of the use of maternal health services [27]. Schools are an indispensable source whereby female youths can learn about reproductive and maternal health care. They are also more likely to have good knowledge on the importance of skilled birth attendance at delivery which makes them more likely to utilize institutional delivery services. Furthermore, educated women are likely to have a better status in their households along with economic independence which enhances their decision-making power to demand, seek and utilize delivery care services [27].

Our study found the likelihood of uptake of institutional delivery increased with an increase in the number of ANC visits. This finding aligns with previous studies [12, 21]. Women are suggested about the benefits of institutional delivery to mother and child and informed about the cash incentives for institutional delivery during ANC visits. However, a qualitative study by Bhandari et al. found that during regular antenatal visits, a significant proportion of women were not clearly advised to seek institutional delivery care and were not given adequate information about the available cash incentives for institutional delivery [19]. Furthermore, a study by Dixit et al. emphasizes that mere antenatal visits do not ensure the reception of appropriate ANC and therefore decent quality and appropriate delivery of information must be ensured to encourage women to uptake the next level of services [40]. It is suggested that when women are satisfied with the content of care they receive during their ANC visits; they are more likely to utilize institutional delivery services.

Education of the household head was not found to be a significant predictor of the place of delivery in our study. Similar findings were observed in another study from Nepal [12]. However, a plethora of studies have determined husband's education as a significant predictor for the utilization of maternal health services [17, 41–43]. While our study could not explicitly explore the role of husbands in institutional delivery uptake, their significance as decision-makers, especially in financial matters, and as companions accompanying women to healthcare facilities should not be underestimated. In certain instances, husbands may be the sole available support for transporting their wives to healthcare institutions for delivery. Consequently, their perception of maternal needs is of critical importance as it can influence delays in seeking maternal health services [44, 45]. Given that husband's involvement directly affects access to and utilization of maternal healthcare services by pregnant women, further investigation of this dimension is imperative in future research, particularly within predominantly patriarchal societies like Nepal.

Women's exposure to mass media was not significantly associated with the place of delivery as well. However, exposure to mass media can promote their utilization of maternal health services as they are an important source of information about health services and their outcomes, which in turn may influence people's decision to seek health care services. Mass media campaigns play a key role in disseminating maternal health care information to mothers having low education level, which is especially useful in the context of South Asia as many mothers have limited formal education or are illiterate. A study among four South Asian countries showed mothers' exposure to mass media increased the likelihood of delivery by a skilled birth attendant by 24–53% [46].

## Policy implications from this study

The findings of this study highlight the persistent inequality in the utilization of institutional delivery services in Nepal, particularly among different socioeconomic and geographical groups. While inequality seems to be on decreasing trend, further efforts are necessary to achieve universal health coverage and eliminate such disparities. The root causes of the health inequity, such as economic status, female education and social discrimination should be addressed in order to address this issue in long term. The burden of out-of-pocket expenditure that poorer women dread could be ameliorated by the expansion of maternal benefit schemes that could compensate out-of-pocket expenditure and opportunity cost to a certain extent in addition to incentives by the Aama program which can be carried out by local level authorities targeting the poorest households in their communities.

## Strength and limitation

This study used the data from a recent nationally representative survey which used the standardized questionnaire, employing a multilevel modelling approach accounting for the hierarchical nesting of complex survey data. We utilized Strengthening the reporting of observational studies in epidemiology Statement to self-evaluate our study. The study tools employed in this study analyzed prior habits and activities retrospectively therefore recall and reporting bias are likely. However, the potential for recall bias were addressed by analyzing the data from the most recent pregnancy within the past 2 years. Service-side determinants like delays in service delivery and quality of care were not considered in the analysis.

## Conclusion and recommendations

Province, area, ethnicity, household wealth, women's education, and the number of ANC visits were significantly associated with institutional delivery. We recommend accelerating the efforts towards attaining universal health coverage. Governmental and other stakeholder's effort must be focused on implementing tailored interventions focusing on illiterate women from rural areas, poorer households, and socially disadvantaged groups. Moreover, we recommend expanding the benefits of maternal benefit schemes targeting the women from the poorest households in the communities.

## Acknowledgments

The authors would like to express their gratitude to Multiple Indicator Cluster Survey for allowing us to access and use the data set for the study.

## Author Contributions

**Conceptualization:** Bipin Thapa.

**Data curation:** Bipin Thapa.

**Formal analysis:** Bipin Thapa.

**Investigation:** Bipin Thapa.

**Methodology:** Bipin Thapa, Anita Karki.

**Software:** Bipin Thapa, Anita Karki.

**Visualization:** Bipin Thapa.

**Writing – original draft:** Bipin Thapa, Anita Karki, Suman Sapkota.

**Writing – review & editing:** Bipin Thapa, Anita Karki, Suman Sapkota, Yifei Hu.

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
