## [Decision Letter · Decision Letter 0]

4 May 2023

PONE-D-23-05202What determines institutional delivery service utilization in Nepal? A multilevel analysisPLOS ONE

Dear Dr. Thapa,

Thank you for submitting your manuscript to PLOS ONE. After careful consideration, we feel that it has merit but does not fully meet PLOS ONE’s publication criteria as it currently stands. Therefore, we invite you to submit a revised version of the manuscript that addresses the points raised during the review process. I echo on the reviewer comments for further improvement of the paper. Please follow the additional comments of editor for further details.  

We look forward to receiving your revised manuscript.

Kind regards,

Kanchan Thapa, MPH, MPhil

Academic Editor

PLOS ONE

Journal Requirements:

Additional Editor Comments:

Dear authors,

I enjoyed reading your paper. It is a really interesting piece of work, however there are rooms for improvement. I echo on the reviewer comments and suggest you to finalize the paper. I have additional comments along with the reviewers’ comments on your paper.

Please also include the roles and academic qualification of each author.

I also suggest to have more review of relevant literature in the field. 

The abstract should be rewritten. It must contain the status of global status of ID, why it is important to research, and why it is important in Nepal’s context. Also, required to brief about the methodology in the abstract section. The use of data source methods, statistical test has been not properly mentioned in the section.

In the conclusion section, you wrote about the role of local level. Have you really relied on the findings of the study for this recommendation?

Background:

Need to have an additional literature review on the subject matter.

What is the SDG target for MMR and how is the progress towards SDG? What are the unfinished agenda of MDGs in Nepal related to maternal and neonatal health?

You mentioned about only a handful of the studies used national level survey data to assess the factors for institutional delivery. Why? What is the research gap on the subject matters?

Methods:

Include the information, how the research was carried out? Secondary or primary data analysis? How the approval was taken from MICS program to use the data for further analysis?

Conceptual framework and expert opinion: what are the theories that you used in the present research to formulate the conceptual framework?

Results:

The whole result section should be rewritten and should present in an analytical way. Only the describing the table won’t be enough. Please review other scientific literature for describing the results.

Have you ever tried to explore the husband role’s for institutional delivery in Nepal? If not, please use all available variable for analysis.

Conclusions and recommendations

Mostly described findings only. Why only the government role’s been discussed, what about the roles of other stakeholder’s role in increasing institutional delivery in Nepal.

Reviewers' comments:

Reviewer's Responses to Questions

**Comments to the Author**

1. Is the manuscript technically sound, and do the data support the conclusions?

Reviewer #1: Yes

Reviewer #2: Yes

2. Has the statistical analysis been performed appropriately and rigorously? 

Reviewer #1: Yes

Reviewer #2: Yes

3. Have the authors made all data underlying the findings in their manuscript fully available?

Reviewer #1: Yes

Reviewer #2: Yes

4. Is the manuscript presented in an intelligible fashion and written in standard English?

Reviewer #1: Yes

Reviewer #2: Yes

5. Review Comments to the Author

Reviewer #1: Review Reports

Title: What determines institutional delivery service utilization in Nepal? A Multilevel Analysis.

Manuscript ID: PONE-D-23-05202.

Reviewer version: RI

Review Comments

1. What are the gains in the reduction of maternal mortality in Nepal and nation specific strategies since 2015?

2. The objective is not SMART.

3. Why systematic sampling while selecting the final HHs?

4. Why don’t you assessed for the socio-economic and cultural factors affecting the institutional delivery in Nepal. What are the exact sources of the conceptual framework?

5. Who collected the data? How was the data collected? What are the measures taken to ensure data quality assurance?

6. Why “The first model consisted 137 of external environment factors. Furthermore, household contextual factors were added in the 138 second model, while individual factors were added in the third model.”?

7. How was the data merged? Did both the data set and source have similar variable? If there was difference, how did you manage it? Is there data that is discarded?

8. Language, grammar and sentences were not written as per the standard of the scientific paper. E.g., A sentence beginning with numeric.

9. The result should be simple, logical and coherent.

10. The discussion and the conclusion should be in line with the objective of the study.

Regards,

Reviewer #2: A well written manuscript. However it will be better if the title is rewritten in the statement instead of the question in alignment with the general objective. otherwise it is okay.

can be accepted.

6. PLOS authors have the option to publish the peer review history of their article (what does this mean?). If published, this will include your full peer review and any attached files.

Reviewer #1: No

Reviewer #2: **Yes: **Richa Acharya

---

## [Author Response · Author response to Decision Letter 0]

10 Jul 2023

Dear editor and reviewers, thank you very much for your suggestions. We have extensively changed the abstract, and main text of the manuscripts as per your suggestions and feedbacks. 

Editor: Please also include the roles and academic qualification of each author. I also suggest to have more review of relevant literature in the field. 

Authors: Thank you very much for your suggestion. We have written the roles and academic qualification of each author while submitting in the submission system. As per your suggestion, we have added more review of the relevant literature in the field, in the manuscript. 

Abstract section: 

Editor: The abstract should be rewritten. It must contain the status of global status of ID, why it is important to research, and why it is important in Nepal’s context. Also, required to brief about the methodology in the abstract section. The use of data source methods, statistical test has been not properly mentioned in the section.

Authors: Thank you very much for your suggestion. We have changed and added lines in the background and methods section of abstract as per your suggestion.

Editor: In the conclusion section, you wrote about the role of local level. Have you really relied on the findings of the study for this recommendation?

Authors: Thank you very much for your suggestion. We have removed the role of local level from the conclusion section of the abstract and retain the part which relies on the findings of our study.

Background:

Editor: Need to have an additional literature review on the subject matter.

What is the SDG target for MMR and how is the progress towards SDG? What are the unfinished agenda of MDGs in Nepal related to maternal and neonatal health?

You mentioned about only a handful of the studies used national level survey data to assess the factors for institutional delivery. Why? What is the research gap on the subject matters?

Authors: Thank you very much for your suggestion. We have addressed all your suggestion in the background section of the manuscript in the from line numbers 55 to 85.

Methods:

Editor: Include the information, how the research was carried out? Secondary or primary data analysis? How the approval was taken from MICS program to use the data for further analysis?

Authors: Thank you very much for your suggestion. We have written “The present study was based on data from the Nepal Multiple Indicator Cluster Survey (MICS) 2019” to indicate the data being used in the study as secondary data. Since, we did not participate in the design of the study and data was download from the MICS site, we have written “The detailed methodology of the survey can be accessed from the Nepal MICS 2019 report. The website of the MICS program makes the anonymized survey dataset freely available under request in line numbers 104 to 106 with references. The approval was taken from MICS program to use the data for further analysis via requesting them in an email.

Editor: Conceptual framework and expert opinion: what are the theories that you used in the present research to formulate the conceptual framework?

Authors: Thank you very much for your suggestion. Conceptual framework was solely based on the literature review and expert opinion. We have clearly mentioned this in line number 120.

Results:

Editor: The whole result section should be rewritten and should present in an analytical way. Only the describing the table won’t be enough. Please review other scientific literature for describing the results.

Authors: Thank you very much for your suggestion. We have reviewed other scientific literatures published in PLOS journal and others as well. We have rewritten some sentences of the result section as per your suggestions.

Editor: Have you ever tried to explore the husband role’s for institutional delivery in Nepal? If not, please use all available variable for analysis.

Authors: Thank you very much for your suggestion. Yes, we tried to explore the husband role in institutional delivery in our study however we could not do it because there is no variable in women data set which could assess the husband role in institutional delivery. Moreover, the man data set which provides variable related to men could not be merged into women data set as there is no key variable given to link these two data sets.

Conclusions and recommendations

Mostly described findings only. Why only the government role’s been discussed, what about the roles of other stakeholder’s role in increasing institutional delivery in Nepal.

Authors: Thank you very much for your suggestion. We have also added the term “stakeholder’s effort” in line number 370 of the conclusion and recommendations section of the manuscript.

Reviewer #1:Review Comments

1. What are the gains in the reduction of maternal mortality in Nepal and nation specific strategies since 2015?

Authors: Thank you so much for you suggestion. We have added the gains in reduction of maternal mortality and nation specific strategies since 2015 in line number 62-74. 

2. The objective is not SMART.

Authors: Thank you so much for you suggestion. We have adjusted the objective to make it SMART.

3. Why systematic sampling while selecting the final HHs?

Authors: The present study was based on data from the Nepal Multiple Indicator Cluster Survey (MICS) 2019 conducted by the Central Bureau of Statistics in collaboration with the United Nations Children’s Fund. We did not participate in the design of the study and data was download from the MICS site. In the Multiple Indicator Cluster Survey (MICS), systematic sampling is often used in the second stage of sampling for selecting households within the selected clusters. This method is employed for practical reasons and to ensure an efficient and representative sample.

4. Why don’t you assessed for the socio-economic and cultural factors affecting the institutional delivery in Nepal. What are the exact sources of the conceptual framework?

Authors: Thank you very much for the question. We have incorporated important socio-economic and cultural factor available in the data sets that affects the institutional delivery in Nepal. We have included religion, ethnicity, wealth status, household head’s sex, age and education, and women’s age and education in our model to determine the potential socio-economic and cultural factors associated with institutional delivery. The conceptual framework was based on the literature review and expert opinion.

5. Who collected the data? How was the data collected? What are the measures taken to ensure data quality assurance?

Authors: Nepal Multiple Indicator Cluster Survey Nepal 2019 team who worked under Central Bureau of Statistics collected the data with the help of trained survey enumerators. They visit selected households and administer structured questionnaires through face-to-face interviews.

MICS utilizes a multipronged approach to data‐quality assurance. Interviewers are extensively trained for an average of three to four weeks prior to conducting fieldwork, with only those mastering the tools being selected. Fieldworkers are organized into teams, with a supervisor who coordinates field activities and revisits households for quality control. During fieldwork, data are transmitted and stored to a cloud server on a daily basis. Every week, survey managers run field‐check tables that measure data‐quality indicators, disaggregated by team and interviewer. This allows corrective action to be taken while interviewers are in the field. 

6. Why “The first model consisted 137 of external environment factors. Furthermore, household contextual factors were added in the 138 second model, while individual factors were added in the third model.”?

Authors: Thank you very much for the question. The model consisted of external environment factors in first model and household contextual factors were added in the second model while individual factors were added in the third model. External environment factors may have a contextual influence on individual outcomes. By including these factors first, we accounted for the contextual variation in the data and estimated the contextual effects on the outcome variables. This helped to capture the impact of the broader environment on individuals within the model.

7. How was the data merged? Did both the data set and source have similar variable? If there was difference, how did you manage it? Is there data that is discarded?

Authors: Thank you very much for the question. The UNICEF MICS website provides information to merge the data set. 

Source:https://mics.unicef.org/files?job=W1siZiIsIjIwMTkvMDQvMDEvMTQvMDAvMjQvNzM0L0ZBUV9NZXJnaW5nX01JQ1NfZGF0YV9maWxlc18yMDE5MDIyOC5kb2N4Il1d&sha=c3117037366d1001

We merged two data sets namely women and household data set. When merging women’s data files with their households, we used the cluster numbers (variable HH1) and household numbers (variable HH2) as key variables available in the both datasets. 

After merging two different data sets i.e., mother and household, a total of 14,805 observations were found. 1,950 observations of married women aged 15-49 years with a live birth in the last two years by place of delivery of the most recent live birth were selected and retrieved. After removing five observations with missing values: the place of delivery (n=1), health insurance (n=3) and household head education (n=1), a total of 1,945 observations remained. Finally, 13 observations which did not fit the criteria for the place of delivery were removed and 1,932 observations were considered in the final analysis.

8. Language, grammar and sentences were not written as per the standard of the scientific paper. E.g., A sentence beginning with numeric.

Authors: Thank you very much for your suggestion. We have checked the language, grammar and sentences again.

9. The result should be simple, logical and coherent.

Authors: Thank you very much for your suggestion. Some of the sentences from the result section have been rewritten in order to make them more simple, logical and coherent.

10. The discussion and the conclusion should be in line with the objective of the study.

Authors: Thank you very much for your suggestion. We have tried to adjust the paragraphs in the discussion and conclusion.

Reviewer #2: A well written manuscript. However, it will be better if the title is rewritten in the statement instead of the question in alignment with the general objective. Otherwise, it is okay.

can be accepted.

Authors: Thank you so much for your suggestion. We have changed the title of the manuscript as “Determinants of Institutional Delivery Service Utilization in Nepal” in statement form in alignment with the objective of the study.

---

## [Decision Letter · Decision Letter 1]

31 Aug 2023

PONE-D-23-05202R1Determinants of Institutional Delivery Service Utlization in NepalPLOS ONE

Dear Dr. Thapa,

Thank you for submitting your manuscript to PLOS ONE. After careful consideration, we feel that it has merit but does not fully meet PLOS ONE’s publication criteria as it currently stands. Therefore, we invite you to submit a revised version of the manuscript that addresses the points raised during the review process.

I am suggesting you to make correction based on reviewer's and academic editor comments as below. After addressing all these comments, please submit it again. 

We look forward to receiving your revised manuscript.

Kind regards,

Kanchan Thapa, MPH, MPhil

Academic Editor

PLOS ONE

Journal Requirements:

Additional Editor Comments:

Dear Authors,

I suggest you to rework, correct or answer on the following aspects:-

Line 122. Please take care of it.

Line 143: Can you please make clear about the line. After the approval of whom, where and how?

Ethical consideration section- you mentioned about ethical consideration and approval from Central Bureau of Statistics. Are they the legal body to provide ethical clearance for your study.

You have mentioned about two heading - Statistical Methods AND data management. Is there any opportunity to make these or three into one. OR is it really required?

While reading, I did not see when the study was conducted? Please mention it.

If you are referring some critical appraisal checklist for your study for self evaluation, it can help you to better improve your study. Therefore, I may suggest you to once refer to such critical appraisal checklist for your study.

Reviewers' comments:

Reviewer's Responses to Questions

**Comments to the Author**

1. If the authors have adequately addressed your comments raised in a previous round of review and you feel that this manuscript is now acceptable for publication, you may indicate that here to bypass the “Comments to the Author” section, enter your conflict of interest statement in the “Confidential to Editor” section, and submit your "Accept" recommendation.

Reviewer #3: All comments have been addressed

Reviewer #4: All comments have been addressed

2. Is the manuscript technically sound, and do the data support the conclusions?

Reviewer #3: Yes

Reviewer #4: Yes

3. Has the statistical analysis been performed appropriately and rigorously? 

Reviewer #3: Yes

Reviewer #4: Yes

4. Have the authors made all data underlying the findings in their manuscript fully available?

Reviewer #3: Yes

Reviewer #4: Yes

5. Is the manuscript presented in an intelligible fashion and written in standard English?

Reviewer #3: Yes

Reviewer #4: Yes

6. Review Comments to the Author

Reviewer #3: It seems like that authors have addressed the raised comments by the editor and the other reviewers, other wise well formulated peace of work. To be a final manuscript i suggest to write more explanation about husdands role-although you mentioned some in the review answer but add little more in the actual text, as this is importnat perspective. Tables need extra explanation and arrangement.

Reviewer #4: Thank you for submitting your research with us, some references need to be updated (References No. 3,4,5,7,9,29,39,44) and all references need editing to be written in the same style.

7. PLOS authors have the option to publish the peer review history of their article (what does this mean?). If published, this will include your full peer review and any attached files.

Reviewer #3: No

Reviewer #4: **Yes: **Fadia Ahmed Abdelkader Reshia

---

## [Author Response · Author response to Decision Letter 1]

6 Sep 2023

Additional Editor Comments:

Editor: Dear Authors, I suggest you to rework, correct or answer on the following aspects: -

Line 122. Please take care of it.

Authors: We have deleted it.

Editor: Line 143: Can you please make clear about the line. After the approval of whom, where and how?

Authors: Thank you so much for the comment. The first author requested access for the MICS Nepal 2019 datasets to the UNICEF MICS Team through their website. Subsequently, the MICS Team granted access to the data via email. Following data access approval, the first author proceeded to download the dataset directly from the MICS website's survey page. We have added few words to make it clearer in in the analysis subsection of the manuscript.

Editor: Ethical consideration section- you mentioned about ethical consideration and approval from Central Bureau of Statistics. Are they the legal body to provide ethical clearance for your study.

Authors: We have not taken ethical clearance from Central Bureau of Statistics for our study. The survey protocol for MICS was approved by the Central Bureau of Statistics as per the Statistical Act (1958) in September 2018. The Statistical Act enables Central Bureau of Statistics to carry out surveys according to the government’s ethics protocol without involving an institutional review board. Our study was based on data from the MICS 2019 conducted by the Central Bureau of Statistics.

Editor: You have mentioned about two heading - Statistical Methods AND data management. Is there any opportunity to make these or three into one. OR is it really required?

Authors: Thank you for the suggestion. We have merged this two sub-heading to one, making it data management and analysis sub-section.

Editor: While reading, I did not see when the study was conducted? Please mention it.

Authors: Thank you for the suggestion. Nepal MICS 2019 was conducted by the Central Bureau of Statistics from May to November 2019 as a part of sixth-round of the global MICS household program, with the technical and financial support from United Nations Children’s Fund Nepal. We have utilized the data obtained from this study. We have added this information in the data source subsection to make it clearer.

Editor: If you are referring some critical appraisal checklist for your study for self-evaluation, it can help you to better improve your study. Therefore, I may suggest you to once refer to such critical appraisal checklist for your study.

Authors: Thank you so much for suggestion. We have used STROBE Statement to self-evaluate our study. We have added this to the strength and limitation section of our study.

Reviewer #3: It seems like that authors have addressed the raised comments by the editor and the other reviewers, otherwise well formulated peace of work. To be a final manuscript i suggest to write more explanation about husbands’ role although you mentioned some in the review answer but add little more in the actual text, as this is important perspective. Tables need extra explanation and arrangement.

Authors: Thank you so much for suggestion. We have added more explanation about husband’s role in the discussion section of our manuscript as per your suggestion. We have added extra explanation for table 3 as per your suggestion.

Reviewer #4: Thank you for submitting your research with us, some references need to be updated (References No. 3,4,5,7,9,29,39,44) and all references need editing to be written in the same style.

Authors: Thank you so much for suggestion. We have updated the references and edited them to be written in the same style.

---

## [Editor Report · Decision Letter 2]

13 Sep 2023

Determinants of Institutional Delivery Service Utilization in Nepal

PONE-D-23-05202R2

Dear Dr. Thapa,

We’re pleased to inform you that your manuscript has been judged scientifically suitable for publication and will be formally accepted for publication once it meets all outstanding technical requirements.

Kind regards,

Kanchan Thapa, MPH, MPhil

Academic Editor

PLOS ONE
---

## [Editor Report · Acceptance letter]

14 Sep 2023

PONE-D-23-05202R2 

Determinants of Institutional Delivery Service Utilization in Nepal 

Dear Dr. Thapa:

I'm pleased to inform you that your manuscript has been deemed suitable for publication in PLOS ONE. Congratulations! Your manuscript is now with our production department. 

Kind regards, 

on behalf of

Mr. Kanchan Thapa 

Academic Editor

PLOS ONE